# Clans, Families and Kinship Structures in Scotland—An Essay

## Bruce Durie

Independent Researcher, Angus, Scotland, UK; bd@brucedurie.co.uk

**Abstract:** Anyone who has visited a Scottish Games or Gathering in North America will be struck by the number of Clan societies occupying tents around the Games ground and participating in a "Parade of Tartans". Yet, a substantial number of these do not represent Highlands or Borders Clans, but are really descendants of Lowland Families. The "Clan" appellation has been applied wrongly to all of Scotland, as though this were the universal or at least the dominant form of social/kinship organization. The cultural appendages of that—kilts, tartans and Gaelic language—are considered uniformly Scottish. In reality, the clan system was a minority social structure in Scotland. The uncritical adoption of the term "Clan" ignores and minimizes the larger and more important Lowland Family structure. The nature of these two structures—Clan and Family—are compared and contrasted, and a case made for greater recognition of the Lowland Family as the pre-eminent form of social structure in Scotland. This has implications for, *inter alia*, genealogy, Scottish cultural and language studies, ethnicity and Y-DNA testing.

**Keywords:** Scotland; Scottish; Scots; clan; family; genealogy; tartan; games; gatherings; ethnicity; DNA testing

## 1. Introduction

In order to get to a discussion of the Scottish Clan, we necessarily have to look back before the time of the flowering of the Clan system (roughly the 14th to 18th Centuries) to see how and why it evolved the way it did. The relevance of all this prefatory material will become clear.

Scotland is not a unitary country in terms of ancient ethnicity. It is a multi-ethnic mosaic of at least five "ancient" lineages (original Pictish-Brythonic, Strathclyde Britons, Irish Gaels, Northumbrian Angles and Norse, with later but smaller overlays from elsewhere). One major implication is that there is no representative "Scottish Y-DNA" but an amalgam of haplogroups still clustered around the original settlement regions. For the purposes of this discussion, "Scotland" is as currently defined—the northern third of the island of Great Britain, plus more than 790 islands[1] principally in the archipelagos of the Hebrides, and the Northern Isles of Orkney and Shetland.

However, the mainland of Scotland has a particular geography that lends itself to becoming a unitary socio-political entity. It is essentially a peninsula, bounded in the south by a long-established Anglo-Scottish border less than 100 miles (160 km) long. This is based largely on the River Tweed, amended by various treaties, conventions and administrative re-arrangements, but its more-or-less present position was established over 1000 years ago, albeit with the occasional inclusion of parts of Cumbria and Northumbria at times, and even (briefly) parts of Yorkshire and Northamptonshire.

Scotland is also divisioned by geo-topography. As described below, the exigencies of plate tectonics and glaciations produced three distinct landscapes: the Highlands and Islands of the north and west which are gloriously mountainous, dominated by glens, river systems and lochs; the flat Lowlands; and the undulating Southern Uplands, more commonly called the Borders area. (Dumfries and Galloway in the extreme south-west is a special case.) This essay will attempt to show how the interplay of incursion, settlement, original ethnicity and geography have produced different social organizations—the Clans

in the Highland and Islands, and a similar structure in the Borders, but not in the Lowlands. We will come to the definition of "Clan" later.

Long before this time, periodic glaciations covering the whole land mass obliterated any signs of human habitation before the Mesolithic period (roughly 15,000–5000 YBP)[2] or during the last interglacial (ca. 130,000–70,000 YBP). It seems the first post-glacial hunter-gatherer groups arrived in Scotland around 12,800 YBP, as the ice sheet retreated. The earliest known artefacts include a flint arrowhead from Islay in the Western Isles, and flints found near Elsrickle, South Lanarkshire. There is evidence of habitation in Tentsmuir, Fife and Cramond, near Edinburgh (both in east-central Lowland Scotland) dating from 10,500 YBP. The earliest known permanent houses on Scottish soil date from around 9500 YBP, and the first villages 6000 YBP, including the well-preserved village of Skara Brae on Orkney.[3] The original inhabitants, undeniably Brythonic, have come to be called "Picts".

In the Iron Age (after the 8th Century BCE)[4] Brythonic ("Pritennic") culture and languages spread into the south-west (Strathclyde), and into south-eastern Scotland, both possibly from Wales (*Cymru*) and Cumbria (*Cwmry*). It remains unclear how much of this was cultural contact rather than invasion, although there was direct settlement.[5] Systems of petty kingdoms developed, large fortified settlements expanded (e.g., Dunbarton, west of Glasgow, and the Votadini stronghold of Traprain Law in East Lothian), considerable numbers of small duns, hill forts and ring forts, and the spectacular brochs were built (Armit 2002; MacKie 1991). Over the course of the first millennium BCE, consolidation of settlement and the concentration of wealth and underground storage of food led to a societal change to a chiefdom model.

The Romans arrived[6] ca. 71 CE but never conquered, and by 142 CE built the Antonine Wall at a natural narrow point, joining the Clyde and Forth rivers. They settled south of that, never conquering the lands to the north but trading, recruiting client tribes and making occasional explorations and punitive expeditions. Their geographers identified the inhabitants living near and just above the wall the as the *Maeatae* and those further north as the *Caledonii*. Later, all the inhabitants north of what the Romans considered *Britannia* were distinguished as Picts, a related group of peoples who spoke a Brythonic (Brittonic) language cognate with Welsh and Cumbric. In any case after only 80 years of never really subduing any Scottish land or peoples, the Romans retreated behind the pre-existing Hadrian's Wall. This became the *de facto* northern edge of the Roman Empire, but is some miles south of the actual Anglo-Scottish border and lies entirely within England (Figure 1).

By the time Roman rule in *Britannia* ended around 410 CE, the various Iron Age tribes native to Scotland had united as the Picts, with Romanized Britons in the southern part, and the Gaelic Irish raiders known to Rome as the *Scotii* settling along the west coast and Western Isles in greater numbers. It is possible that three groups coalesced in the Great Conspiracy (*barbarica conspiratio*) (Frend 1992) that overran Roman Britain in 367 CE along with the mysterious Attacotti, and sea-borne Saxons from Germania. If so, it argues for regular contact and communications between all these groups. Rome did leave two lasting and related legacies—literacy and Christianity, both of which arrived in Scotland courtesy of Irish missionaries, and the earliest historical accounts of the native "Scots" appeared.[7]

There things stood until the Picto-Gaelic Kingdom of Alba was forged by King Alpin (*Alpín mac Echdach*) and his son Kenneth I (*Cináed mac Ailpin*), King of Dál Riada (841–850), King of the Picts and the King of Alba (843–858). Kenneth I, who was probably Pict himself, cohered the Gaels and Picts, largely to fend of various Norse ("Viking", see below) incursions starting at that time.

Northumbria was annexed and Lothian granted to Scotland in 973. Scottish control of the Lothians (in the east) was settled by the Battle of Carham in 1018 and the Solway—Tweed line was legally established in 1237 by the Treaty of York between England and Scotland.[8] This is the border today, except for the "Debatable Lands" north of Carlisle, and

a small area around Berwick-upon-Tweed, which was taken by England in 1482. Berwick was not fully annexed into England until 1746, by the Wales and Berwick Act 1746.[9]

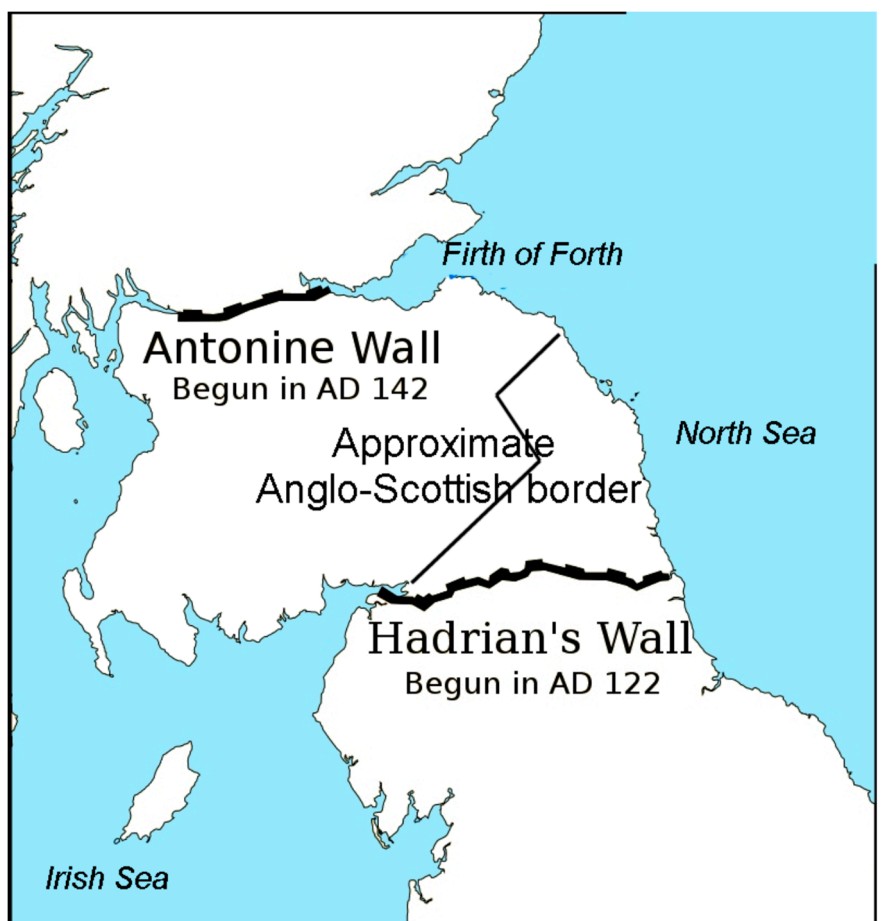

**Figure 1.** The position of the Antonine wall (wholly within Scotland), Hadrian's Wall (wholly within England) and the approximate Scotland-England border (which changed at various times).

Now turning to the west, during the 5th and 6th Centuries CE the Irish Gaels consolidated Dál Riata in present-day Argyll (*Earra-Ghàidheal*, meaning border or coast of the Gaels) in Scotland and part of Co. Antrim in Ulster (now in Northern Ireland). Gaelic language (Goidelic)[10] and culture, originating in Ireland, was eventually dominant throughout the rest of Scotland and the Isle of Man. This may have much to do with the influence of Gaelic writing and ecclesiastical dominance, although the Picts of Kenneth's time (9th Century) were socially and politically dominant.

There were three more major incursions that added to Scotland's ethnic diversity.

Bernicia was the kingdom established by Anglian settlers in the 6th Century, extending from the Forth to the Tees, approximating to the modern north-east English Northumberland, Tyne and Wear, Durham, and southeastern Scottish Berwickshire (in the eastern Borders) and East Lothian, and bordered to the west by the Brythonic kingdom of Strathclyde. In the early 7th Century, Bernicia merged with its southern neighbour, Deira, to form the kingdom of Northumbria, which at time switched between Scottish and English rule. Although an Anglo-Saxon kingdom, and undoubtedly bringing the Germanic influences of Old English that would in time forge the related languages of Northumbrian and Scots, it is noticeable that the local name for the peoples was Gododdin (possibly a derivative of the Latin Votadini, see above); the 6th Century epic poem in (Brythonic) Welsh, *Y Gododdin*, was actually written just south of Edinburgh.[11]

Next came the Norse, popularly but wrongly named Vikings. The word possibly derives from Old Norse *víking* meaning a journey or voyage, *víkingr* a rower on such a journey,[12] and was never indicative of a nationality or region in Scandinavia. It is not a nationality or ethnic group so much as a job-description. In any case, the Scandinavian raiders did not comprise a unitary ethnic or genetic group. In Old English, the word *wicing* came to signify a sea-raider or pirate, for obvious reasons. The Irish called them *Dubgail* and *Finngail* ("dark and fair foreigners", giving the lie to the myth that they were all flaxen haired), the Gaels *Lochlannaich* ("people from the land of lakes"), the Anglo-Saxons *Dene* (Danes) and the Frisians *Northmonn*. The bulk of the raids came from the areas around the Kattegat and Skagerakk sea areas. The present day nations of Norway, Sweden and Denmark did not exist then as we understand them now, but as a shorthand, we could say that "Swedes" and southern "Norwegians" tended to invade and settle along the north and west coasts of Scotland,[13] the Isles, and in Ireland from Dublin to as far south as Cork; whereas eastern Scotland (and England) mainly received incursions from "Danes", although that is not a universal rule (Figure 2a). Icelanders were originally "Norwegian".

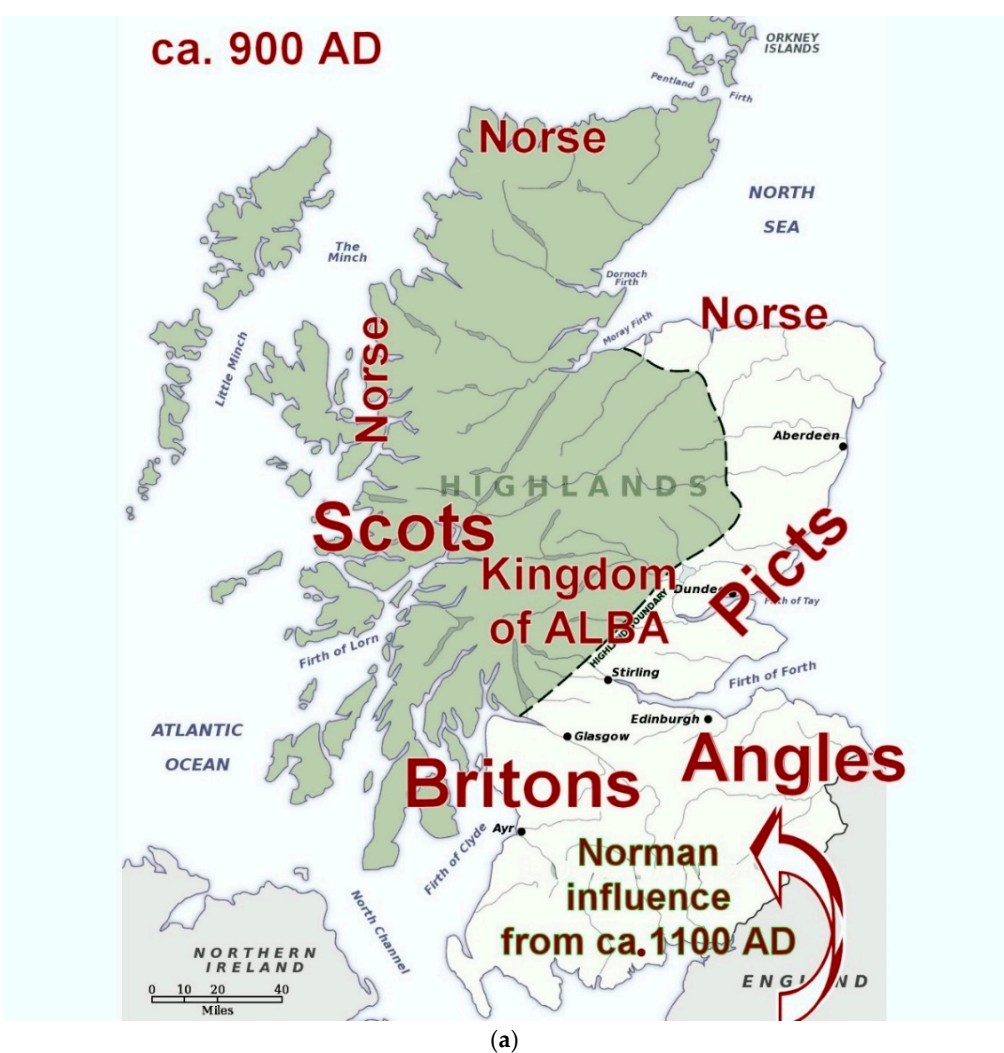

(a)

**Figure 2.** *Cont.*

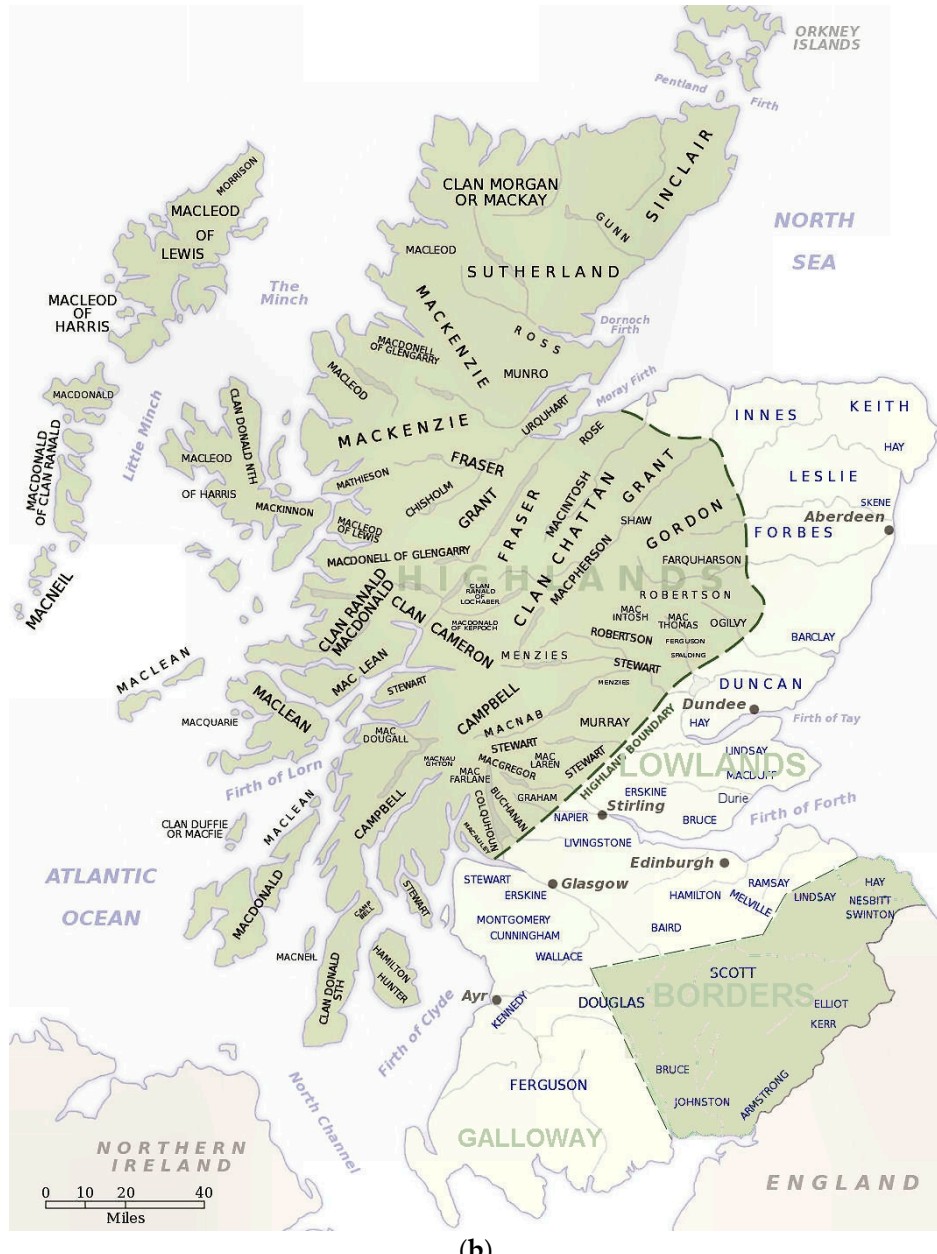

**(b)**

**Figure 2.** (**a**) The various ethnic incursions into Scotland at various times. (**b**) An approximate Clan and Family map, showing the geo-topographical division of Scotland into Highland and Islands (north and west), Lowlands (central and north-east), Borders (south and east) and Galloway (south-west). All such surname maps are a snapshot in time, and not complete, but give a rough indication of where Clans and Families consider their "heartlands" to be.

## 1.1. Anglo-Normans

Thus, what the Romans had found was an admixture of Picts and Strathclyders, both Brythonic. The next millennium saw the arrival of Goidelic Gaels and Bernician Angles, welded together by the Alpin dynasty. Then, came the Norsemen followed in the 12th and 13th Centuries by Anglo-Normans. All of these had different social organisations initially, and predominated in different parts of the country—as a shorthand: the Gaels were in the west and north Highlands and Islands, although later Gaels to Galloway in the south-west; the remains of the Picts were in the eastern Lowlands; Britons were in Strathclyde and the Lothians; Angles were also in the Lothians and the eastern Borders; The Norse were mainly in Caithness, Orkney and Shetland and the western coastal fringes and Isles, although



some Danes had occupied parts of the eastern coast. The south and east of Scotland was shifting linguistically to Scots, related to northern English. However, by the 11th Century, the royal blood was no longer like that of the Alpin dynasty, Gael and Pict, but was more Dane and Saxon. Malcolm III Canmore's mother was a daughter of Siward the Dane, Earl of Northumbria; he married Margaret, sister of the Saxon Edgar Ætheling; their son David I, progenitor of the later kings of Scotland, married the heiress of Siward's son Waltheof. The west was Gael in character, the north was Norse, the east was increasingly Teutonised.

The Normans never conquered or ruled Scotland. However, there were two waves of Normanised Anglos and Saxons who were granted lands in Scotland, mostly in the Lowlands. The first were those who arrived in the wake of Malcolm's dynastic and military alliance with Edgar Ætheling. One telling of this is that Malcolm, early in his reign, travelled to the court of Edward the Confessor in 1059 to arrange a marriage with Edward's kinswoman Margaret, who had arrived in England two years before from Hungary (Duncan 2002). The marriage did not take place then, which may explain why in 1061 the Scots invaded Northumbria, where Margaret's family had settled, and plundered Lindisfarne.[14] Alternatively, Malcolm and Edgar allied against William the Conqueror's harrying of the north in the winter of 1069–1070, where the presence of Edgar Ætheling, last Wessex claimant to the English throne, had encouraged Anglo-Danish rebellions, and at that point a strategic marriage to Margaret was arranged in 1070 (Keene 2013). Either way, Margaret became Queen of Scots in 1070 and proceeded to have at least seven children, three of whom later became kings, plus a bishop, a queen (Matilda, of England) and a Countess (Mary, of Boulogne).

An unreliable list of the "illustrious exiles" who accompanied Edgar and Malcolm would include some well-known Scottish surnames: Borthwick, Lesley, Lindsay, Livingston, Maurice, Maxwell and three brothers named Melville, plus Gospatric, ancestor of a number of landed and noble families (Henderson 1879). Contrary to popular telling, none of these was "Hungarian".

Margaret brought a number of other things to Scotland, including some refinement, Middle English as a language, the Roman Catholic form of worship to replace the Culdee church, and the influence of the Norman court (which Malcolm shared). Malcolm Canmore had begun to emulate Anglo-Saxon England by creating Earls and Barons as marks of distinction to those who had helped him overthrow Macbeth, and he continued this practice with those who fled north from William's rule, or were part of Edgar's Saxon and Northumbrian revolt.

More Anglo-Normans and Bretons came with or in the wake of Malcolm and Margaret's son, David I (ca. 1084–24 May 1153, Prince of the Cumbrians from 1113 to 1124, reigned as King of Scots from 1124). This had an immense influence on the political, societal and land-holding aspects of Scottish life. David had been exiled to England temporarily in 1093 and lived at the court of King Henry I after 1100. Henry was married to David's sister, Maud or Matilda, so David and was also Henry's nephew. David was given the hand in marriage of a different Maud or Matilda, Countess of Huntingdon (ca. 1074–1130/1131), great-niece of William the Conqueror and daughter of Waltheof, Earl of Northumberland and the last of the great Anglo-Saxon earls who retained power after the Norman conquest of England in 1066. This brought David the immense wealth of the Huntingdon Earldom, which title stayed in his line until 1237.

David had seen the value of the feudal system, both political and financial, under which powerful local magnates could have heritable possession of and control over large swathes of land, but under conditions of fealty and homage payment to the king. This allowed David to turn many disputatious and potentially rebellious northern Chiefs and Mormaers ("Earls") into vassals—but whose regional authority was now backed by the throne—and to make extensive land grants in the southern part of the country to his friends and supporters, under similar conditions. These include Bernard/Burnett, Wallace, Walter fitz Alan (ancestor of the Stewarts/Stuarts) and others. In such a way did the ancestors

of Robert Bruce—already transplanted from Cherbourg into Yorkshire having followed Henry I after his victory at Tinchebray in 1106–come to be Lords of Annandale.

This relatively small number of Anglo-Normans made very little impact on the genetics of Scotland, but dominated at least the Lowlands and Borders in what was now becoming the Scottish Feudal baronage and peerage. Their descendants are among the highest-titled (Earls, Marquesses and Dukes) in Scotland, and remain some of the major landholders. They also became the Bruce and Stewart/Stuart kings. David I also organised Church and State unity and extended Anglo-Norman institutions.

The Gael resentfulness towards the monarchy—regarded as "Anglo" and not speaking Gaelic—set up a long series of conflicts over the centuries. It also solidified the differences between societal organisation: the Clans in the Highlands and the Families in the Lowlands.

### 1.2. The Clan System and Its Definition

Modern sociologists and anthropologists used the term "clan" in a particular way—typically characterized by assumed or actual kinship and descent, often claiming descent from an apical ancestor, founding member or patriarch, and tending to be endogamous (clan members can marry one another) but not exclusively so. However, this essay is not an exploration of how the Scottish clan fits with a post hoc sociological definition; that definition itself is based on the existence of clans in Scotland, then applied to other societies, present and past. It would be elliptical to take the modern academic definition of "clan" as applied to other societies and cultures, and use that to examine whether or not Scotland had "clans" as defined. The word "clan" as a descriptive label for the organization of society in the Scottish Highlands and Ireland entered English in the early 15th Century. The etymological origin may have been *plant* as found in Middle Welsh and cognate with the Latin *planta* (offshoot, sprout, cutting), but as the Goidelic languages have no initial p this is usually substituted by k or hard-c.

Strangely, although the word *clann* in Irish signified progeny, none of the Gaelic words for kinship groups is equivalent to "clan". These include:

- *teaghlach* (nuclear family, or extended family in the same household);
- *muintir* ("kinsfolk" in the broad sense);
- *líon tí* (either "family" in the sense of "household", or everyone in the household including non-relatives);
- *fine* (the closest equivalent to the English meaning of "clan" (Dónaill 1992).

However, in Scotland the word "clan" had a subtly different connotation—a territorial/kinship/quasi-military structure, largely confined to the Highlands and Islands, although a similar structure can be applied to the eastern Borders (Figure 2b).

The reason for the clan structure is geo-topological. The economy of the Irish clans had been pastoral—lush, green pastures and arable land, in a countryside that was both flatter and more temperate than the sparse Highland moors and hillsides away from the coasts as found in Scotland. It also offered less by way of natural defences than the Scottish glens. This meant that the spread-out "kindred" was a more natural form of organisation in Ireland. The Highlands of Scotland largely consist of glens (valleys) broadening out from a point where a river enters along a *strath* (a flat river valley with steep hills on both sides) to end at some larger body of water—a wider river, an inland loch, a sea-loch, or the sea itself. The meant a defensible defile, fertile but limited land for pasturing cattle and some small agriculture—whatever grains would grow on the marginal hillsides, and seasonal common grazing of livestock—access to fish and transport, and ample opportunities for both contact and conflict with the inhabitants of neighbouring glens. The individual glens, straths and lochsides were looked after by lesser "gentry" of the clan. Naturally, some of these would band together under an acknowledged Chief and engage in cattle-raiding, hostage-taking and outright warfare against neighbouring emergent clans. However, they could unite against a common enemy or threat, and bonds of amity were reinforced by hostage-taking, fosterage and the exchange of wards, and so forth. In such ways, great confederations like the Clan Chattan were forged. This system took naturally to feudalism (see below).

Many clans claimed (and still claim) fabled founders, often based on Irish mythology, plus largely fictitious king-lists and ancestor-lists, which tended to differ depending on who compiled them and for what purpose. This romantic and glorified sentiment of origin reinforced a quasi-royal status, and to this day gives the impression that everyone with that clan surname is descended from the patriarch. For instance, Clan Donald is often claimed to descend either from Conn (a 2nd Century king of Ulster), or Cuchulainn, the legendary hero of Ulster. Their traditional and political enemies Clan Campbell have claimed their progenitor to be Diarmaid the Boar, from the Fingalian or Fenian Cycle. Clans Mackinnon, Clan Gregor and MacNab are among those placed within the Siol Alpin group (claiming a common descent from Alpin, father of Kenneth MacAlpin, who united the Scottish kingdom in 843, as above). Modern historical scholarship and the disruptive technology of DNA testing (below) has challenged most of this. In fact, only one confederation of clans (including Sweeney, Lamont, MacLea, MacLachlan and MacNeill), can trace their ancestry back to Niall of the Nine Hostages, the 5th Century High King of Ireland. The progenitors of most clans cannot be authenticated further back than the 11th Century, and in most cases there is no continuity of lineage until the 13th or 14th Centuries.

The clan system as we now know it probably developed after the Battle of Bannockburn in 1314–there is no mention of the word "clan" in any of the chronicles describing the Wars of Independence fought by Robert Bruce at that time. The historic clans were small. They could field in battle only a few hundred men, suggesting a total clan membership in the region of 500–1500. Modern clans, with considerable input from Scottish descendants overseas can number in their thousands.[15]

The emergence of clans as we now know them had more to do with politics than ethnicity or origin. When the Scottish Crown conquered Norse-dominated Argyll and the Outer Hebrides in the 13th Century, and the Mormaer of Moray and the northern rebellions of the 12th and 13th Centuries were pacified, there was an opportunity for local war-lords to dominate nearby families who accepted their protection, and often the clan name. These warrior chiefs were in origin Gael, Norse-Gael and British. During the Wars of Scottish Independence of that period, King Robert Bruce used the idea of feudal tenures introduced by David I to control the powers of clans—he granted charters for land and recognised chief as Barons (reinforcing their local powers, but under the King)–in exchange for political and military support in the national cause against the English. Clan Donald, for example, was elevated in status above Clan Dougall, although they were of common descent from the great 12th Century Norse-Gaelic warlord Somerled. Clanship was thus not only a strong tie of local kinship but also of feudalism to the Scottish Crown, reinforced by Scots law. However, the pattern of "small" folk joining in with a powerful clan for protection and land to plant, giving in return homage, fealty and a fighting arm, and adopting the patronymic, means that surname (as we now understand it) is no guide to genetic origin. Famously, the Inverness Frasers offered a boll (six bushels) of meal to anyone who would adopt the name.

The original concept of heritage bound up with the clan was not surname. Well into the 13th Century, surnames (in the sense of passing unchanged from fathers to sons) were a rarity. Fergus, Iain's son (MacIain or Johnson) would have sons all called Fergus's son (MacFergus or Fergusson). The chief was the patriarch, head, main landowner, defender, military commander and dispenser of justice, surrounded by his immediate "aristocracy" or *derbhfine*, and his *fine* (the clan's warrior elite, not necessarily related). He might have had in his household a physician (often hereditary, but of a different clan, such as the MacBeaths), a *shennachie* (the historian, genealogist and keeper of the memory of the clan, again often a role fulfilled from outwith the clan, but heritably, such as the MacMhuirichs), a piper, a steward or butler, and so forth.

Although formally a system of ultimate land-ownership by one person or a small group of truly related individuals, the *ethos* was one of shared stewardship of land, to be defended, extended and passed on to the next generations. Everyone in the clan felt a commonality of possession and would fight to the death for it. In truth, the Highlands evolved two distinct but correlated ideas of heritage. First, there was the collective heritage

of the clan, known as their *dùthchas*. This was the prescriptive right to settle within the lands where the chiefs and leading members of the clan gave customary protection, recognition of the personal authority of the chief and leading clan members as trustees for the clan and their lands. Second, was the *oighreachd*, (or *eiraght*, meaning "heritage" in the sense of stewardship and inheritance, over and above mere ancestry), the acceptance that charters granted by the Crown and other powerful land owners to the chiefs, chieftains and lairds defined the limits of the estate settled by the clan—this gave a legal authority to the clan chief and leading *derbhfine* as landed proprietors, who held the land heritably in their own right, and had certain rights and responsibilities, including judicial and baronial. These were largely removed in 1746.[16]

By the 14th Century, there had been further influx of ethnicities from Norman or Anglo-Norman and Flemish roots, such as the clans Cameron, Fraser, Menzies, Chisholm and Grant. They folded in with the local customs and structures.

*1.3. The Lowlands*

The emergent social structure in the Lowlands was different, but also a consequence of geography. Large, flat and fertile plains in the central belt and along the north-east coast were suitable for settled agriculture, and crown grants of large swathes of land to one baron or noble within the feudal system meant that the relationship between the ultimate landholder and the smaller landholders, tacksmen (lease-holders) tenants and agricultural labourers was purely economic rather than "of blood". That said, a number of people in a given area might come to share a surname, but again, this was not necessarily an indication of a genetic link—names might be locational, but also occupational (Baxter meaning baker, Webster meaning weaver, Ferrier meaning iron-worker, etc.), moreso than in the Highlands where there were far fewer specialised functions.

The coincident growth of Royal Burghs and Burghs of Barony further concentrated the artisans, merchants and administrators, and also helped consolidate the economic, political and legislative power of the local magnates. The Lowlands were closer to the capital and royal palaces, and the influence of the Crown was more immediate than in the Highlands.

A labourer could move to another employer, a tenant could go elsewhere at the end of a tenancy or lease, a small landholder could sell up and move, younger sons could seek land of their own and not always nearby. People could move to towns. In short, there was not the same bond of place, fealty and (perceived) kinship as in the Highlands.

However, the Lowland landowners and heads of Families were often noble, and also Feudal Barons. The basic feudal concept is of a hierarchy of heritable possession—all land ultimately belonged to the Crown, but was granted in feu to tenants-in-chief, termed Barons. The payback (*reddendo*) was originally a stated amount of military service by so many armed men, but eventually collapsed into payment in cash or kind—farm produce, fish, wood for fuel, etc. Barons could sub-infeudate (parcel out or sell, heritably) parts of their estate to others, whether family or not, again in return for service or payment of some kind, even if only nominal ("peppercorn"). That meant the Lowland lairds had the same legal status as enshrined in the Highland *oighreachd*, with judicial powers, etc. These were also blunted by the 1746 Heritable Jurisdictions (Scotland) Act and, perhaps as an unintended consequence, flattened the distinction between Lowland Chief and Highland Chief (who in any case were becoming more like their Lowland counterparts in manners, tastes and lifestyles). Many castles were abandoned. Some chiefs moved to live in a more refined way in Inverness, Edinburgh or even London. Gaelic had dwindled away as the predominant language in the Highlands, just as it had in the Lowlands centuries earlier. However, although the Battle of Culloden in 1746 is often seen as the watershed moment for the Clan System, in truth it was in decline long before that.

*1.4. The 1587 Act against Clans and the 1609 Statutes of Iona*

The first mention of "clans" in any Act of the Scottish Parliament or other statute is in 1587 (often mis-dated as 1597)[17]

> "held at Edinburgh upon 29 July 1587 for the quieting and keeping in obedience of the disorderly persons, inhabitants of the borders, highlands and isles"

It contained a description of the

> "Chiftanis and chieffis of all clannis . . . duelland in the hielands or bordouris"

and, helpfully, a listing of these. Noticeably, the Lowlands are not included, but are mentioned separately three times, to distinguish the families there from the Highlands and Borders clans (Table 1).

This mention of "Chiftanis and chieffis of all clannis . . . duelland in the hielands or bordouris" is often mistakenly applied as if "Highlands and Borders" encompassed the whole of Scotland, and that therefore the term "clan" equally refers to Lowland families.[18] In reality, it specifically excludes the Lowlands, such as (in Fife alone) Bruce, Lindsay, Leslie, Durie, Hay and others, whose chiefs were among the most influential men in Scotland at the time.

A Lowland family, moreover, may well have had an armigerous chief and feudally held lands, but did not have the typical clan structures. Clans are therefore a phenomenon of the Highlands and Borders and the equivalent kinship/territorial structure in the Lowlands is the Family, usually based on a feudal barony.

There is one earlier mention, in 1384, of a "clan", which is a special case—legislation of Robert II, which enjoins "the lord earl of Fife . . . as head of the law of Clan MacDuff" to "protect the present statute and ordinance". This is Robert Stewart, Earl of Fife until 1420, an illegitimate son of Robert II, Duke of Albany, Governor of Scotland from 1406 and Regent to three Scottish monarchs: Robert II, Robert III, and James I. The leadership of Clan MacDuff (clann meic Duibh) was subsidiary to the position of Mormaer or Earl of Fife, and an example of the chief's surname not being that of the clan or family. The chief of MacDuff was not always the Mormaer, especially after feudal primogeniture was applied to the mormaerdom in the reign of Duncan I (1133–1154)–the head of the clan, Macduff of Fife died leading the common soldiers of Fife at the Battle of Falkirk (22 July 1298) rather than the Mormaer, Duncan IV of Fife, who was a minor (b. 1289) and last male Gaelic ruler of Fife.

Various records of Parliament and of the Privy Council from the 16th and 17th Centuries use the word "clan" to denote surname-based alliances, usually tempestuous, and in order to quell them in some way. The Statutes of Iona, passed in 1609, banned hospitality, strong drink, the sheltering of fugitives, the existence of bards, and other aspects of the traditional culture. They also required Highland Chiefs to have their heirs educated in Scots-speaking or English-speaking Protestant schools in the Lowlands. Some clans, such as the MacDonalds of Sleat and the MacLeods of Harris, adopted the Reformed religion, while others, including the MacDonalds of Clanranald, Glengarry, Glencoe and Keppoch, remained Catholic.

During the Cromwellian or Commonwealth period (1642–1660) there was increasing oppression of Scotland as a whole, but especially of the Highlands. The Hanoverian era (1714–1837), covers the two major Jacobite Risings of 1715 and 1745–there were others.

The Disarming Act of 1715 did not have the desired effect. The Battle of Culloden in 1746 was the final straw for a British government determined to crush the power of the Clans and tame Scotland—although there were more Scots fighting at Culloden against Bonnie Prince Charlie than on his side.

The Abolition of Hereditary Jurisdictions Act of 1747, discussed above, removed most of the Chief's legal and administrative powers over his clan as a Baron. The proscription against wearing tartan (unless in a Highland regiment of the British army, such as the Black Watch) greatly diminished the prominence of the clans.

**Table 1.** The list of Clans in the Highlands and Borders felt to be "disorderly".

| The roll of the clannis that hes capitanes, cheiffis and chiftanes quhome on thai depend, oftymes aganis the willis of thair landislordis, alsweill on the bordouris as hielandes, and of sum speciale personis of branches of the saidis clannis | The roll of the clans that have captains, chiefs and chieftains whom on they depend, often times against the will of their landlords, as well on the borders as highlands, and of some special persons of branches of the said clans |
| --- | --- |
| [The Borders] | [The Borders] |
| Middle Marche | Middle March |
| • Armestrangis<br>• Crosaris<br>• Ellottis<br>• Niksonis | • Armstrong<br>• Crosier<br>• Elliott<br>• Nixon |
| West Marche | West March |
| • Batesonis<br>• Bellis<br>• Carruthers<br>• Glenduningis<br>• Grahmes<br>• Irwingis<br>• Jardanes<br>• Johnestonis<br>• Latimeris<br>• Litillis<br>• Moffettis<br>• Scottis of Ewisdaill<br>• Thomesonis | • Bateson<br>• Bell<br>• Carruther<br>• Glendinning<br>• Graham<br>• Irving<br>• Jardine<br>• Johnston<br>• Latimer<br>• Little<br>• Moffat<br>• Scotts of Ewesdal<br>• Thomson |
| Hielandis and Iles | Highlands and Isles |
| • Clanandreis<br>• Buchananis<br>• Clanchamroun<br>• Campbellis of Innerraw<br>• Campbellis of Lochnell<br>• Clanquhattan<br>• Clandonoquhy in Athoill and partis adjacent<br>• Clandowill of Lorne<br>• Fergussonis<br>• Fraseris<br>• Grahmes of Menteth<br>• Grantis<br>• Clangregour<br>• Clan Jeane<br>• Clankanze<br>• Clankynnon<br>• Clanlawren<br>• Clanlewyd of Harray<br>• Clanlewis of the Lewis<br>• MacFerlanis, Arroquhar<br>• Makintoscheis in Athoill<br>• Clan MacThomas in Glensche<br>• Clane MacKane of Avricht<br>• MacKnabbis<br>• Menzess in Athoill and Apnadull<br>• Monrois<br>• Murrayis in Sutherland<br>• Clanneill<br>• Clanrannald in Loquhaber<br>• Clanrannald of Knoydert, Modert and Glen Gardy<br>• Spaldingis<br>• Stewartis of Athoill and partis adjacent<br>• Stewartis of Buchquhidder<br>• Stewartis of Lorne or of Appin | • Clan Andrew<br>• Buchanan<br>• Clan Cameron<br>• Campbell of Inverawe<br>• Campbell of Lochnell<br>• Clan Chattan<br>• Clan Donachie in Atholl and parts adjacent<br>• Clan Dowell of Lorne<br>• Ferguson<br>• Fraser<br>• Graham of Menteith<br>• Grant<br>• Clan Gregor<br>• Clan Ian<br>• Clan Kenzie<br>• Clan Kinnon<br>• Clan Laren<br>• Clan Leod of Harris<br>• Clan Lewis of the Lewis<br>• MacFarlane, Arrochar<br>• MacIntosh in Atholl<br>• Clan MacKean of Ardvorlich<br>• MacNab<br>• Clan MacThomas in Glenshee<br>• Menzies in Atholl and Apnadull<br>• Munro<br>• Murray in Sutherland<br>• Clan Neil<br>• Clan Ranald in Lochaber<br>• Clan Ranald of Knoydart, Moidart and Glengarry<br>• Spalding<br>• Stewart of Balquhidder<br>• Stewart of Lorne or of Appin<br>• Stewarts of Atholl and parts adjacent |

*1.5. The Reinvention of Highland Culture in the 19th Century*

This justifies a study all on its own,[19] but in essence there was a revival of interest in all things Highland, fostered by the Highland Society of London. This body, founded in 1778 for Highland gentry living in England, started the revival of Gaelic heritage, culture, language and identity. The Society did many fine things, but perpetrated a few errors in its enthusiasm The Society later recognised that the kilt and tartan were in danger of becoming archaic, and in 1815 started the collection of samples of tartans which 'authenticated' by various chiefs of clans as being distinctive to their particular clan. This started the inaccurate one-to-one identification of surname with tartan.[20]

This was polished to a higher gleam by Sir Walter Scott's stage-managing in 1822 of a visit by King George IV to Scotland which more or less required the Clan Chiefs to 'rediscover' their 'ancient' tartans, which then became fashionable, rather than regarded as the dress of barbarians, in polite society. Queen Victoria later fell in love with the Highlands and bought and largely rebuilt Balmoral Castle as she thought a 'traditional' Scottish castle should look, and Prince Albert wore a kilt in the Balmoral Tartan that he (a German) had designed, as he admired the 'noble, warrior race' symbolism of the Highlander.

Then, there was a breath-taking fraud perpetrated by the two 'Sobieski–Stuart' brothers (actually Welshmen called Allan) who never actually claimed, but allowed it to be thought, that they descended from the Royal Stuarts via Prince Charles Edward (Bonnie Prince Charlie) in Poland. They said they had found—but could never actually produce—an ancient book called the *Vestiarium Scoticum* ("The Dress of Scotland") depicting the ancient tartans of the clans. Walter Scott himself was scathing, and a highly critical anonymous review appeared in the *Quarterly Review* now known to have been authored by the unimpeachable Rev. Dr. Mackay, editor of the Highland Society's *Gaelic Dictionary* and George Skene, then professor of History at the University Glasgow (Stewart et al. 1980).

All of this led ineluctably to the growth of the Highland Games and Gatherings movement, which in reality had it finest flowering in North America, then spread back to Scotland O (with a few notable exceptions, such as the first "modern" Highland games held in 1781 at Falkirk, which is firmly in the Lowlands. It also fomented the syllogism Scotland = Highlands = kilts and tartans (Mitchell 2005; Trevor-Roper 1983)).

*1.6. The Borders*

The Act of 1587 (above) took pains to distinguish the Highlands and Borders from the Lowlands. The word "clan" applies to this region too, and the in habitants were just as unruly and not amenable to control by the authorities in Scotland or England. Some surnames straddled the actual border and largely disregarded it, to the fury of monarchs in both countries. Thus a Borders Bell or Graham or Carruthers might be considered Scottish or English, partly depending on where the actual border was at any given time, and partly out of personal wish or sentiment. Famously, Border Reivers operated along the Anglo-Scottish border from the late 13th Century to the beginning of the 17th. They included Scots and English, and they raided the entire border country without regard to their victims' nationality. Their heyday was in the last hundred years of their existence, during the time of the Stuarts in Scotland and the Tudors in England.

It is easy to see why this situation existed in the Borders. In the late Middle Ages, Scotland and England were frequently at war, and the lives and livelihoods of those living near the Border were ravaged by the contending armies advancing or retreating. Even when there was no formal war, tensions remained high, and royal authority on either side was often weak, especially in remote areas. Communities and kindreds found security in group strength, and loyalty to a far-away monarch meant law and governance had little effectiveness. In addition, a family that straddled the borders had to deal with a system of partible inheritance is evident in some parts of the English side of the Borders but primogeniture on the Scottish side. Land might be divided equally amongst all sons (in England) or by the eldest (in Scotland), leading to a situation where the inheriting generation might have land too small for survival, or none at all. What developed was a

predatory way of living in parts of the Borders. The geography consists of low, undulating hills or open moorland, good for grazing cattle and later sheep but not for arable farming. Livestock was easily rustled from the open hillsides and Reivers also stole household goods or valuables and took prisoners for ransom, impartially but usually where those raided were not under the protection of a powerful noble or landholder.

The attitudes of the authorities either side of the border wobbled between indifference and even encouragement at times, and at others oppressive and desultory punishment. The families found themselves considered the first line of defence against cross-border invasion (in either direction) or an intolerable nuisance. They were also used to their lands being devastated by advancing and retreating armies, regardless of allegiance. It is no wonder they took to lawlessness, which became intolerable to the two governments, and to a clan-like structure of mutual help and defence against other clans. They built fortified tower houses and Peel towers for defence and warning-signalling, were more loyal to clans than to nations, saw everyone else as a source of plunder, and lived in a state of constant alert and preparedness. It is said that at one point the Armstrongs alone could put 3000 men on horseback with a day, far more and far faster than any King or Queen could muster. It is not surprising then that Borderers were in demand as mercenary soldiers and light cavalry, served with Scottish or English armies in the Low Countries and Ireland (sometimes as an alternative to official penalties) and as levied soldiers the Moss Troopers were crucial in the battles of Flodden (1513) and Solway Moss (1542). However, they were uncontrollable even by the local March Wardens (who were often Borderers themselves, and not averse to taking sides). Queen Elizabeth I of England is quoted as saying that "with ten thousand such men, James VI could shake any throne in Europe".

When Elizabeth died, the situation along the border was such that the English Parliament considered rebuilding and re-fortifying Hadrian's Wall, which runs south of the "Debatable Lands" of the Borders (Fraser 1989). Upon his accession to the English throne, James VI and I acted against the reivers with harsh "justice", abolished border law, and dropped the very term "Borders" in favour of "Middle Shires", exiled some of their leaders to Ireland and so forth. He and his successors passed various laws to quell the Borders and their Clans.[21]

In summary, while there have been attempts to find a suitable term to describe the structure of Borders families, such as the "Riding Surnames" or "Graynes" (James 1986), they can be likened to the Clan system of the Highland and Islands. The Scotts of Buccleuch and of Harden and others displayed the archetypical features of: patriarchal leadership by the Chief of the name; defined territories in which most of the kinship lived, with a defendable main seat; systems of fosterage, and also of tutorship when an heir to the chiefship was a minor; oaths of fealty; bonds of manrent; and so forth. They are, in effect Borders Clans.

However, most of this went away in the aftermath of the Union of Parliaments in 1707.

## 2. The Ulster Plantations

This will be necessarily brief, as it does not impinge on the discussion of what a Clan was or is, but may serve to remind readers by one of the routes by which Scots (mainly Lowlanders in this case) reached the Americas.

The organised colonisation (plantation) of Ulster—the northernmost province of Ireland—began during James VI & I's reign. Small privately funded plantations by wealthy landowners began in 1606, while the official plantation began in 1609 (Falls 1996; Perceval-Maxwell 1999). Most of the land thus colonised had been confiscated from native Irish Gaelic chiefs and nobles, several of whom had fled Ireland for mainland Europe in 1607 in the aftermath of the Nine Years' War,[22] also called Tyrone's Rebellion. Most of the settlers (or planters) came from southern Scotland and northern England, were predominantly Protestant and Scots-speaking, and had a culture and customs which differed in other ways from those of the native Irish who felt, not surprisingly, dispossessed. They became the Ulster-Scots (wrongly called the Scots-Irish or Scotch-Irish in America and Canada).[23]

This put them cheek by jowl with the descendants of the Gaels who had migrated from precisely the same area to the west of Scotland a thousand years earlier. Bear in mind that these western Scottish Goidelic Gaels had intermingled with the Pictish and other Brythonic Scots, especially as the interstices—Caithness, western Aberdeenshire, Perth, Stirlingshire, western Angus, the border between Argyll and Ayrshire, and in the already part-Gael-part-Brythonic-part-Anglo area of Dumfries and Galloway in the south-west. In addition, the Stuart kings had purposely sent artisans and merchants to specially created Royal Burghs in the Highlands and Islands (in particular Kirkwall in Orkney, Rothesay on Bute, Cromarty, Wick and later Campbeltown), partly in order to improve the local economies and also to strengthen royal control via client earls and other nobles. There was some migration of Highland Gaels to Ulster (mainly from adjacent Argyll), but in the main it was Lowlanders and those from the western Borders shires.

This had led to a great of genealogical confusion, particularly in North America, where some descendants of these peoples regard themselves as Irish and some as Scots, which they further confuse by adopting the garments and trappings of the Scottish Highlands and Clans, against all historical logic.

## 3. Migration from Ireland to Scotland

A particularly ham-fisted piece of legislation in the time of Queen Anne, aimed at blunting the influence and wealth of Irish Roman Catholics, had the unintended consequence of affecting Ulster (Scots) Presbyterians as well.[24] Many thousands emigrated to the Colonies (Caribbean, West Indies and North America) but some retrenched back in Scotland, where they may still have had links. In the absence of documentation, it is often impossible to trace whether a family called Montgomery, say, originated in Ayrshire and stayed there, or went to plant Ulster but returned to Ayrshire. The situation is even worse for Kennedy—there is an Ayrshire-origin branch which also went to Ulster, but also a southern Irish branch, completely unrelated, and scions of both of these may have turned up in the same place in America at the same time.

The situation became even more complicated with the mass-migrations for Ireland during the famines, and in the light of the Industrial Revolution. From the 1830s onwards, a perfect storm of wars, famines, unemployment, poverty and religio-political persecution caused thousands to migrate to Scotland from Ireland (and also Italy and Eastern Europe, both Jewish and Catholic). Between 1830 and 1914, census data show that over 300,000 Irish people migrated to Scotland.[25] In the main, they settled in the West of Scotland, particularly Glasgow and Lanarkshire. Besides the attraction of proximity, this area was booming with new industries, the building of canals, railways and roads and ships. Other industrial centres were Ayrshire, Dundee and Paisley, whereas Edinburgh, Perth and Stirling figured less due to the relative lack of employment.

This had the effect of pouring back into Scotland the genetic "cousins" of the original Gaels, but not in the Highlands and Islands. However, at the same time, Highlanders were also heading for the large industrial conurbations for precisely the same reasons.

### 3.1. Is There an Official List of Clans and Families?

Now we come to one of the main reasons that many Scots abroad undertake genealogy, DNA testing and other investigations; they want to know "Which Clan am I in?"

Sadly, there is no official list (see Table 1).[26] Any of the bodies which could draw up such a thing have steadfastly resisted doing so, seeing the inevitable tensions in Scotland, but mainly in North America and other places where Scots have settled, and look back to an imagined past with a misty eye and an imperfect knowledge of history. Many surname groups have set up membership bodies with names like Clan X Society and Clan Y Association, when X and Y are Lowland surnames. There is seen to be some value or status in being a "Clan" rather than a "Family", which derives from a romanticized vision of what Scotland was and is, everyone rushing around the glens and straths in a kilt (of a named tartan) brandishing a claymore. These groups resist being told that their ancestry is firmly

Lowland, and refuse to concede this even when shown on a map. The extreme position is "everyone is Scotland is in a Clan" is no more sensible than "everyone in America is in a Tribe", and asserting that the whole of Scotland wore kilts and tartans and spoke Gaelic is as nonsensical as claiming everyone in Germany wore lederhosen, or the whole of England indulges in Morris Dancing.

There is a great deal of emotional capital invested in being part of "our Clan", not to mention the financial investment in the costume and accessories. However, sense is starting to prevail. The Chief of Bruce, Andrew Bruce, earl of Elgin and Kincardine, has been successful in making sure none of the Bruce surname groups use the word "Clan"–they now have names like "Family of Bruce International"–yet the Clan Crawford Association (and many others) continue with that usage.

*3.2. Emigration from Scotland to North America*

This is not the place for an in-depth discussion of emigrants at various times and the routes taken. Suffice is to say that:

- the majority of the Scottish population was in the Lowlands, rather than the Highlands and Borders (Table 2);
- direct emigration from the Lowlands, plus Lowland Scots-derived Ulster, vastly outnumbered that from the Highlands;[27]
- much is written about the Highland Clearances—while these were devastating and to be decried, the Lowland Clearances took place over a longer time, involved more people, and obliterated an entire stratum of society—the cottar—but are less written about, possibly because the emigrants came singly or in small family groups rather than entire communities or ship-loads, as in the Highlands (Aitchison and Cassell 2003);
- at a conservative estimate, only one-third of Scots-ancestry Americans will have had any historical connection to a Highland Clan;[28] individuals' descriptions of their Scottish ancestors' travels to North America often involve some "narrative of victimhood" (we were cleared off our land/religiously persecuted/transported as criminals/sold into servitude, etc.)–while these events undoubtedly happened, they are numerically far less significant than is commonly assumed, and the main reason for emigration was economic betterment and the raising of rents and food prices in their homeland (Durie 2018).

**Table 2.** The populations of counties, then summed into the regions Highlands (H), Lowland (L), Borders (B) and Galloway (G). Webster's original survey (not exactly a census) asked Church of Scotland ministers in 909 parishes to provide various numbers, including the total number of inhabitants. Figures for 1801 and 1811 are from the official government censuses. * Perth and Stirling are considered 50% Highland and 50% Lowland, but the regional totals have been analysed with and without this assumption. Either way, the Highlands represent about one-third of the Scottish population and the Lowlands over half. The counties of Galloway are a special case, neither Highland, Lowland nor Borders.

| H/L/B/G | County | Webster 1755 | 1801 | 1811 | | Webster 1755 | % | 1801 | % | 1811 | % |
|---|---|---|---|---|---|---|---|---|---|---|---|
| H/L/B/G | County | 1,265,380 | 1,608,420 | 1,805,864 | *Assuming Perth and Stirling are half Highlands, half Lowlands* | | | | | | |
| H | Argyll | 66,286 | 81,277 | 86,541 | | | % | | % | | % |
| H | Bute | 7125 | 11,701 | 12,033 | Highlands | 376,901 | 31 | 437,744 | 30 | 462,639 | 28 |
| H | Caithness | 22,215 | 22,609 | 23,149 | Lowlands | 691,293 | 57 | 857,085 | 58 | 1,001,362 | 60 |
| H | Inverness | 59,563 | 72,672 | 77,671 | Borders | 69,820 | 6 | 77,450 | 5 | 83,947 | 5 |
| H | Moray | 30,604 | 27,700 | 27,967 | Galloway | 76,859 | 6 | 106,726 | 7 | 117,385 | 7 |
| H | Nairn | 5694 | 8322 | 8406 | | | | | | | |
| H | Orkney | 23,381 | 24,445 | 23,238 | *Assuming Perth and Stirling are entirely in the Highlands* | | | | | | |
| H | Ross & Cromarty | 48,084 | 50,318 | 60,853 | | | % | | % | | % |
| H | Sutherland | 20,174 | 28,117 | 23,629 | Highlands | 455,466 | 37 | 525,948 | 36 | 558,876 | 34 |

**Table 2.** *Cont.*

|  |  | Webster 1755 | 1801 | 1811 |  | Webster 1755 | % | 1801 | % | 1811 | % |
|---|---|---|---|---|---|---|---|---|---|---|---|
| H | Zetland/Shetland | 15,210 | 22,379 | 22,915 | Lowlands | 612,728 | 50 | 768,881 | 52 | 905,125 | 54 |
| H/L | Perth * | 120,116 | 125,583 | 134,300 | Borders | 69,820 | 6 | 77,450 | 5 | 83,947 | 5 |
| H/L | Stirling * | 37,014 | 50,825 | 58,174 | Galloway | 76,859 | 6 | 106,726 | 7 | 117,385 | 7 |
| L | Aberdeen | 116,168 | 122,100 | 133,871 |  |  |  |  |  |  |  |
| L | Angus | 68,883 | 99,053 | 107,187 |  |  |  |  |  |  |  |
| L | Ayr | 50,000 | 84,207 | 103,839 |  |  |  |  |  |  |  |
| L | Banff | 38,478 | 37,216 | 38,433 |  |  |  |  |  |  |  |
| L | Clackmannan | 9003 | 10,858 | 12,010 |  |  |  |  |  |  |  |
| L | Dunbarton | 13,837 | 20,710 | 24,189 |  |  |  |  |  |  |  |
| L | East Lothian | 29,709 | 29,986 | 31,050 |  |  |  |  |  |  |  |
| L | Fife | 81,570 | 93,143 | 101,272 |  |  |  |  |  |  |  |
| L | Kincardine | 23,057 | 20,349 | 27,439 |  |  |  |  |  |  |  |
| L | Kinross | 4889 | 6725 | 7245 |  |  |  |  |  |  |  |
| L | Lanark | 81,726 | 147,692 | 191,291 |  |  |  |  |  |  |  |
| L | Midlothian | 90,412 | 122,597 | 148,607 |  |  |  |  |  |  |  |
| L | Renfrew | 26,645 | 78,501 | 93,112 |  |  |  |  |  |  |  |
| L | West Lothian | 16,829 | 17,844 | 19,451 |  |  |  |  |  |  |  |
| B | Berwick | 23,087 | 30,206 | 30,893 |  |  |  |  |  |  |  |
| B | Peebles | 8008 | 8735 | 9935 |  |  |  |  |  |  |  |
| B | Roxburgh | 34,704 | 33,121 | 37,230 |  |  |  |  |  |  |  |
| B | Selkirk | 4021 | 5388 | 5889 |  |  |  |  |  |  |  |
| G+ | Dumfries | 39,188 | 54,597 | 62,900 |  |  |  |  |  |  |  |
| G | Kirkcudbright | 21,205 | 29,211 | 33,684 |  |  |  |  |  |  |  |
| G | Wigtown | 16,466 | 22,918 | 20,801 |  |  |  |  |  |  |  |

These figures are taken from James Gray Kyd. Scottish population statistics, including Webster's Analysis of population, 1755. Edinburgh, Published for the Scottish History Society (1975) by Scottish Academic Press, originally published Edinburgh: T. and A. Constable, 1952. Available to view at https://digital.nls.uk/125885863 (accessed on 3 November 2022).

*3.3. Implications for Y-DNA*

The significance of all this is that there is no "Scottish Y-DNA". A simplistic view would put Gaels in the north and west (and Islands) but intermingled with Norse; Gaels and Picto-Scots in the Lowland west; Picts in the east and south central areas; Angles mixed with Picts in the eastern Borders, Gaels mixed with Picts and "Welsh" in the south-west, and Norse in the north and Orkney. Each of those ethnicities would have had its own Y-DNA signatures, but they may not have been unitary (not all Norse are genetically similar, for example).

It is woefully common for men who have their Y-DNA tested to proclaim an origin for which they have no actual evidence. Some examples from the author's own recent professional practice:[29]

- the Sutherland who proudly proclaimed his origins as being "from Sutherland" without any actual evidence, and whose Y-DNA firmly places his origin in the south-west;
- four Grahams, only one of which could be identified with typical "Borders Graham" or "Montrose Graham" Y-DNA signatures (the derivation of which are themselves controversial);
- a Morrison who was adamant about an origin in Uist, but whose profile clearly aligned with unrelated Morrisons elsewhere (it is one of many such surnames of multiple and independent origin);
- a Sinclair who was convinced all Sinclairs came from Orkney or Caithness, but in fact that is a very small branch of one of two other Sinclair lines from East Lothian, and whose ancestors had demonstrably lived in the West of Scotland in the 1800s;
- an entire group of people surnamed Black who are convinced to the point of shouting that they are Irish when all the documentary evidence shows them to be originally from Scotland, albeit with a genetic signature that suggests a Gael origin, and with a descendant who may well have lived in Ireland briefly before emigrating to North America.

There are good and cogent reasons for all this.

First, surnames were not really fixed until the 1600s. Even then, they might be occupational (Baxter, Webster, Dewar, Smith) and therefore not typical or any one area in particular, or taken from a physical characteristic—Short, Little, Broad, Buie/Bowie (fair-haired) Dubh/Duff (dark), Bain (fair) and so on.

Next, especially in the Highlands, someone moving onto the land of a powerful chief or major landholder might naturally assume the appropriate surname—partly for reasons of protection or to express fealty, but also just because it was expected that everyone in the chief's "tail" (following) and living in the same place would have the same surname.

Then, people did genuinely move around, individually or in large groups. The Keiths were offered land in the north-east if they would swap it with the king for their lands in the south of Scotland, so their destination became known as (and still is) Keith. Some of the Mackays of Ross fled northwards into Strathnaver in Sutherland, a situation confused by at least three origin stories that has them originally descended from Forbes or Farquharson, or possibly both, or perhaps the early rulers (Mormaers) of Moray. There are many more examples.

Furthermore, surnames have changed over time. Some Gaelic names have been "anglicized", for example: MacEachern to Mckechnie; Mac Gille Bride to Kilbride (and not necessarily from any place called Kilbride); Mac na Aonghus to MacInnes/MacInnis (and no relation to Innes); Mac dubh sith to both MacDuffy and MacFie (but no relation to MacDuff); and so on. The entire surname MacGregor (Clan Gregor) was proscribed in 1603 and many chose other surnames including Gregory, Murray, White, MacGeorge and others—the infamous Rob Roy MacGregor (1671–1734) was actually baptized with his mother's name, Campbell, and bore it most of his life until he decided to get involved in the Jacobite risings of 1689, 1715 and 1719 (Tranter [1991] 2005).

All that said, there are still good reasons to at least attempt to match a surname, a geographical location and a Y-DNA signature. However, this is hampered by the absence of a decent all-Scotland Y-DNA benchmarking survey,[30] persistent wishful thinking by those who do take such tests as to their origins, and the fact that, in the main, Scots themselves do not take Y-DNA tests. Therefore, it is often left to Scots descendants in other parts of the globe to do so, and to try to derive a "paper" genealogy or match to someone who has. Unfortunately, many such online genealogies are merely copied uncritically from other online genealogies which themselves are suspect, demonstrably wrong, shot through with guesswork, and so on.

Some take the most basic tests—Y 37-marker, for instance—which may produce large-scale agglomerations such as "R1b" and "I" are not helpful, as they may mask divergences many tens of thousands of years ago. The Big Y-700 offered by Family Tree DNA is becoming the standard.[31]

*3.4. Autosomal DNA*

While Y-DNA testing, and the reporting on the results, is generally robust, the whole field of autosomal DNA is no better than "genetic astrology"[32] in the phrase of Dr Adam Rutherford, a geneticist at University College, London and a regular BBC presenter. Furthermore, summaries of autosomal DNA testing such as "40% Scottish, 40% Irish, 20% Scandinavian", or even less helpfully "80% British (Scots, English, Irish, Welsh)" are completely bogus. The samples on which these are based tend to an amalgam of self-identified "ancestral origin" claims by the testees (which may not be accurate, or are no more than wishful thinking) or are taken from modern populations, whose ethnicity profiles are not guaranteed to be the same as predominated in those regions centuries before. Equally, they rely on modern borders which may have no relevance to an earlier time. For example, a Russian-speaking Ashkenazy Jew from an area then considered in Poland but now in Lithuania may be co-located with a German-speaking ethnic Slav, and Germany did not exist as a separate and unified nation until 1871 (Haardt 2017). In the British context, it is difficult to define the actual origin of a Y-chromosome that initially migrated to Scotland

from Ireland in the 6th Century CE, then moved back to Ulster in the 17th Century, and re-migrated back to Scotland in the 19th Century—is this "Irish" or "Scottish"? It is certainly not "British", in that it not derived ultimately from a Brythonic origin, but will also be found widespread across the (now) British Isles.

Over the years, the author has asked various genetic testing companies what proportion of their "ethnicity assignments" are derived from: (a) ancient DNA samples (e.g., from archaeological sites or burials); (b) modern-day population sampling (and how extensive); (c) testees self-declared "origins". In all cases the answers were evasive, circumlocutory and/or shrouded in claims of "commercial sensitivity"[33] (personal communications, various).

## 4. Conclusions

The social structures of the Highland and Borders Clan and the Lowlands Family were fundamentally different. The desire of many Scots descendants overseas to be part of a "clan" is at odds with the history and statistics of emigration, and surname membership organisations often adopt the form "Clan X Society" when there was no such Clan in Scotland. In any case, the ancient clan system—which had persisted from the late 13th or early 14th Century—was almost gone before 1746 and the Battle of Culloden, and its 19th Century Romantic reinvention was less than authentic.

The use of Y-DNA to localise surnames is proving valuable, but is hampered by lack of benchmarks in Scotland. More Scots who can prove an ancestral location ca. 1800 should be encouraged to test.

**Funding:** This work was supported in part by the award to BD of US-UK Fulbright Senior Studies Scholar 2015–16.

**Data Availability Statement:** Not applicable.

**Conflicts of Interest:** The author declares no conflict of interest. The funding sponsors had no role in the collection, analyses, or interpretation of data; in the writing of the manuscript, and in the decision to publish the results.

## Notes

1  See "Scottish Executive Resources" (PDF). *Scotland in Short*. Scottish Executive. 17 February 2007. https://www.gov.scot/Resource/Doc/923/0010669.pdf (accessed on 22 July 2022).

2  YBP = years before present.

3  At that time, Scotland was covered in forests and bog-land, and the main form of transport was by water. However, in the Northern Isles and Western Isles, the lack of trees led to building in local stone, and hence Neolithic habitation, ritual, defensive and burial sites are especially well-preserved.

4  BCE = before common era.

5  It is possible that any groups of people in Britain who identified as 'Britons' (*Pritani*) called themselves by something similar to *Cum-ri*, from *combrogi*, signifying 'fellow countrymen'. It is instructive that the English called the Welsh *Wēalas* which may be descended from *walha* (foreigner, stranger) itself a rendition of Latin *Volcae*, a generic term for non-Germanic tribes in the Western Roman Empire, which also survives in the place-names Cornwall, Walloon, Wallachia (Romania) and Welschschweiz (French-speaking Switzerland), the surnames Walsh, Welch and Wallace, and possibly even the in Old French *galeis, galois, gualeis* (Gauls), whence the Modern French *gaulois*. See Faull (1975).

6  Most of present-day Scotland was never under Roman political or military control. See (Hornblower et al. 2012).

7  The scholar-monk Bede of the twinned monasteries of Monkwearmouth and Jarrow in Northumbria, who wrote his *Historia ecclesiastica gentis Anglorum* (An Ecclesiastical History of the English People) in the years before 731, claimed the Picts came from "Scythia", but meaning Scandinavia.

8  "Scotland Conquered, 1174–1296". https://www.nationalarchives.gov.uk/utk/scotland/conquered.htm The National Archives. Last accessed on 22 June 2022.

9  Wales and Berwick Act 1746 (20 Geo. II, c. 42).

10  Incidentally, the word 'gaelic' is pronounced gay-lic in Ireland, but gah-lik in Scotland, reflecting the actual pronunciations in Irish (*Gaeilge*) and Scottish (*Gàidhlig*)–plus the Manx language (*Gaelg*) of the Isle of Man.



11  *Gododdin* may be a variant of *Votadini*. The name is recorded as *Votadini* in classical sources, as *Otodini* on maps of Roman Britain, as the early medieval kingdom called *Guotodin* in Old Welsh as, and in later Welsh as *Gododdin*—see Stuart (1982).

12  "Viking" is a fairly modern name for the Norse sea-raiders. It may derive from an Old Norse term related to a long-distance sea journey, from the same root as Old Norse *vika*, a 'sea mile', the distance after which a new shift of rowers would take over to give the others a rest. ee Beard, David. "The Term "Viking"". archeurope.com. *Archaeology in Europe*. Archived from the original on 7 April 2012. Retrieved 23 July 2022.

13  It does explain, however, why the most northerly coast of Scotland is called Sutherland—it was south as far as the Norse were concerned, and under the rule of the *Jarl* (Chieftain, but cognate with Earl) of Orkney. The Norse also named what is now called the Pentland Firth (*Petlandsfjörð*, the fjord of the Picts, although it is neither a firth nor a fjord but a strait)—see Anderson ([1893] 1990). Before the Norse occupation of Orkney, this strait was known as the Sea of the Orcs, the reference being to the Pictish tribe who inhabited Orkney. Perhaps J. R. R. Tolkien borrowed the name.

14  Edgar, along with sisters Margaret and Christina, and their mother Agatha, were living in Northumbria, as far from William as possible.

15  The word "diaspora" is deprecated. A diaspora is the result of intentional explusion, exile or dispersion of a defined group, first applied to Moravian protestants in 1825 and to Eastern European Jews from 1869; the etymology is Greek *diaspora* from *diaspeirein* ("to scatter about". Scots were not forcibly expelled *en masse* from Scotland, but overwhelmingly left voluntarily albeit often as economic migrants. A better word would be "exodus", although it was not one single migration of a connected group of people.

16  The Heritable Jurisdictions (Scotland) Act 1746 (20 Geo. II c. 43) was passed in the aftermath of the fifth Jacobite rising of 1745–46, and removed the judicial rights which were a considerable fount of the power of Clan Chiefs over their tenants. But it also affected the Lowland landholders. The long title makes this clear: *An Act for taking away and abolishing the Heretable Jurisdictions in Scotland; and for making Satisfaction to the Proprietors thereof; and for restoring such Jurisdictions to the Crown; and for making more effectual Provision for the Administration of Justice throughout that Part of the United Kingdom, by the King's Courts and Judges there; . . . and for rendering the Union of the Two Kingdoms more complete. For remedying the inconveniences that have arisen and may arise from the multiplicity and extent of heretable jurisdictions in Scotland, for making satisfaction to the proprietors thereof, for restoring to the crown the powers of jurisdiction originally and properly belonging thereto, according to the constitution, and for extending the influence, benefit, and protection of the King's laws and courts of justice to all his Majesty's subjects in Scotland, and for rendering the union more complete.*

17  1587, 8 July, Edinburgh, Parliament, see *Parliamentary Register*, 29 July 1587.

18  For a well-known example, see Sir Crispin Agnew of Lochnaw, *Clans, Families and Septs*, www.electricscotland.com/webclans/clans_families_septs.htm, 13th August 2001 (accessed on 4 May 2013).

19  For more on this, see Durie (2018).

20  There are often variants of one tartan, such as 'ancient', weathered, hunting, dress, etc.

21  1606 (4 Jas. 1. c. 1), *An act for the utter abolition of all memory of hostility . . . ; 1609* (7 Jas. 1 c. 1) dealing with criminal law in the region; 1662 (13 & 14 Cha. 2. c. 22), Moss Troopers Act; and a continuation of that in 1666 (18 Cha. 2 c. 3), which stipulated that "notorious thieves and spoil-takers in Northumberland or Cumberland" were to be transported to America, "there to remaine and not to returne".

22  Not to be confused with the later Nine Years' War of 1688–1697, often called the War of the Grand Alliance or the War of the League of Augsburg, between France and a coalition of other European powers.

23  The term "Scots-Irish" was originally coined in America as a pejorative, to give the misleading impression that these were just more Irish, and therefore Catholic, destitute and "other".

24  The *Sacramental Test Act* of 1704–This was aimed at suppressing Catholicism in Ireland, but had the unintended consequence of forcing Presbyterians to take the sacrament of the Lord's Supper according to the rites of the (Anglican) Church of Ireland as a condition of holding any civil or military office under the Crown. This was a major stimulus for emigrations from Ulster to North America.

25  Census data from the General Register Office for Scotland—see *Census Records at the National Records of Scotland*, https://www.nrscotland.gov.uk/research/guides/census-records (last accessed 20 July 2022).

26  The assertions in this section are based on decades of observation, conversations and personal communications, but everyone contacted in relation to this article declined to be named or quoted. The whole issue is just that sensitive.

27  Most of the Scottish settlers to America who came prior to 1854 came from the region of Glasgow, Lanark, Renfrew, Ayr (21.7%) Edinburgh and Lothians (10.6%); Argyll (13.9%), Inverness (9.3%) and Perthshire (8.7%); the Southwest (8.9%). Conservatively, that means 32.3% from the Lowlands, 31.9% from the Highlands. See Myra Vanderpool Gormley (Gormley 1989), Magazine of American Genealogy; Datatrace Systems, Stephenville, Texas, Vol. 4, No. 1; Datatrace Systems, Stephenville, Texas. See also a summary at https://www.genealogymagazine.com/scottish-migration/ (accessed on 1 June 2022).

28  See Note 27 above.

29  References cannot be given for reasons of client confidentiality privacy, and justified General Data Protection Regulation (GDPR) concerns.

30  There have been autosomal surveys, but the Y-DNA signatures cannot be extracted from the data.

31   https://www.ftdna.exe (accessed on 1 June 2022) and specifically https://www.familytreedna.com/products/y-dna (accessed on 1 June 2022).

32   Dr. Adam Rutherford. See, for example, the Genetic Literacy Project (https://geneticliteracyproject.org/2015/06/09/whos-your-ancestral-daddy-family-tree-genetics-might-link-everyone-to-king-david/ (accessed on 1 June 2022).

33   Likewise, these companies cannot be named for legal reasons.

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
