# Peer review of "Clans, Families and Kinship Structures in Scotland—An Essay"

_genealogy, doi:10.3390/genealogy6040088_

Round 1

Reviewer 1 Report

The article is about everything, from the prehistory of Scotland to the identity of modern Scots. An extensive introduction, dedicated to the ancient and early medieval times is overly developed. Considerations about clans and their specifics take up a smaller part of the article. A large part of the text refers to the place of clans in the modern Scottish mentality. Perhaps the title and the text should be altered, emphasize the contemporary tradition of the clans?

Author Response

This was purposely subtitled "an essay" and was intended to be discursive.

It it not possible to understand the context in which Clans and Families operate without considering the origins, historical geography, etc.

I'm happy to consider a modification to the title.

BD

Reviewer 2 Report

This is a wide ranging article, rooted in the established literature for clanship and extending it in the ways claimed. I would make three observations by way of asking for improvement. First, it takes an awfully long time - 6+ pages - to get to the important stuff/contribution - and I think that some clearer sense right at the start of why this extended opening discussion is needed would help readers; Second, the readability of the article is in place poor. Short paragraphs (one is just one line long) make this very difficult to read and fractures the argument; finally, right at the end we get two sections extending the discussion to Ireland, the first dealing with the plantation and the second linking scotland and Ireland through the mechanism of migration, Neither for me work; the plantation section is short and deals in only the most superficial way with the established literature; the migration section deals only in outline with a context of migration literature which is substantial indeed. So these sections either need to be longer and more clearly rooted in the literature or shorter and more summative.

Author Response

I'm grateful to this reviewer.

  1. I have improved the beginning, with additional context.
  2. No apology for short sentences or paragraphs in places - this is usually a device of emphasis. However, the copy-editors may choose to revise this.
  3. I have added to the the Plantation and Migration sections marginally, but these are not really germane to the argument. Signposting is given to the literature if anyone is especially interested. To explore these topics the way they deserve is would turn this essay into a book!

Reviewer 3 Report

The authors address an important topic that is very appropriate for the journal Genealogy and should be of interest to many of your readers.  I am not well-versed with the specific geographic area being examined, but the general historical background information appears to be sufficient.  I am familiar with the larger theoretical issues concerning human kinship in general and clans in particular. I think the paper has potential, but I would like the following issues addressed before publication.

1)      A number of unusual typos/errors need correcting.  I think they might be due to final edits taking place AFTER proofreading.  These include, but may not be limited to, the following:

p. 3 “ended in Britannia ended around” Rewrite and delete one of the “ended”s.

p. 4 “not a nationality of ethnic group” Authors probably meant to write “not a nationality or ethnic group.”

p. 4 Authors probably want to delete one of the two “job description” phrases.

p. 9 “there lands” should be “their lands”

p. 9 “1745-36” appears to be inaccurate.

p. 16 should “which clan am in?” be “which clan am I in”?

2)      The paper would benefit from an introductory paragraph that helps the reader know which of following historical details are most relevant to the main argument. Their first sentence (Scotland is not a unitary country in terms of ancient ethnicity.” Is great. However, I would like to see a couple of following sentences explaining exactly how this connects to their thesis.

3)      The paper would benefit greatly from being placed within a less parochial perspective.  The word “Clan” has been used for centuries by anthropologists around the world to describe a fundamental aspect of human social behavior. Thus, it is crucial to know exactly what a clan is, before having to evaluate the authors’ thesis about the extent to which clans did or did not exist at various times in various areas of Scotland. There should also be a clear overview of the worldwide existence of clans in the ethnographic record, and a description of how an actual clan comes to exist. This discussion should stress the fact that actual clans are typically unilineal and exogamous. Thus, clans are seldom if ever coterminous with residential groups, AND clan names allow people to identify many members of other clans as kin (issues central to the anthropology of social organism since Durkheim). Readers will be in a much better position to evaluate the authors’ arguments about Scotland once they have a clear understanding of what a clan is, how clans come to exist, and why they have come to exist. It is impossible to adequately explain the tendency to make false claims about the existence of clans without first explaining why actual clans have been such an important part of human social behavior.

To clarify the meaning of the word “clan” as anthropologists have used it, I suggest the authors examine the following source:

Palmer, C. T., and L. B. Steadman. 1997. Human kinship as a descendant-leaving strategy: a solution to an evolutionary puzzle. Journal of Social and Evolutionary Systems 20(1):39–51 [especially pages 43-46].

Author Response

I am grateful to this Reviewer.

  1. Various typos and other solecisms (however they arose) have been corrected.
  2. The Intro has been strengthened.
  3. There is no point in a wider discussion of where and/or how the Scottish Clan fits with the modern sociological/anthropological use of the word "clan". The word and the concept were initially borrowed from the Scottish word (in existence from the early 1400s) and societal structure with its main flowering from the 14th to the mid-18th Centuries. So, examining the Scottish Clan in the light of what modern anthropologists consider a "clan" to be, or because the word "clan" has been applied (in English) to Somali qabiili, Norse ætter, Romani vista, sub-tribes of Native Americans, Post-Soviet criminal "clans", and various distinct social groupings in China, Korea, Japan and elsewhere, would be elliptical and tautologous. No Clan Chief in the 17th Century Highlands sat down with a book by Durkheim written ca. 1910 and pondered whether or not the word "clan" was justified.
    That said, I have expanded the discussion on etymology and alternatives on (new Page 7).